

# Forecasting stock indices with the COVID-19 infection rate as an exogenous variable

Mohammad Saha A. Patwary[1] and Kumer Pial Das[2]

[1] Mathematical Sciences, Butler University, Indianapolis, IN, United States of America
[2] Research, Innovation, and Economic Development, University of Louisiana at Lafayette, Lafayette, LA, United States of America

## ABSTRACT

Forecasting stock market indices is challenging because stock prices are usually nonlinear and non-stationary. COVID-19 has had a significant impact on stock market volatility, which makes forecasting more challenging. Since the number of confirmed cases significantly impacted the stock price index; hence, it has been considered a covariate in this analysis. The primary focus of this study is to address the challenge of forecasting volatile stock indices during Covid-19 by employing time series analysis. In particular, the goal is to find the best method to predict future stock price indices in relation to the number of COVID-19 infection rates. In this study, the effect of covariates has been analyzed for three stock indices: S & P 500, Morgan Stanley Capital International (MSCI) world stock index, and the Chicago Board Options Exchange (CBOE) Volatility Index (VIX). Results show that parametric approaches can be good forecasting models for the S & P 500 index and the VIX index. On the other hand, a random walk model can be adopted to forecast the MSCI index. Moreover, among the three random walk forecasting methods for the MSCI index, the naïve method provides the best forecasting model.

## INTRODUCTION

The world has struggled and passed through one or more pandemics almost every century. All pandemics affect the world and make it vulnerable to all extents, including but not limited to the health, social, and economic system. In the past 100 years or so, the world has been affected by the pandemics such as the Spanish flu in 1918, the Asian flu in 1957, the Hong Kong flu in 1968, and the Swine flu in 2009. World equity markets have experienced a turbulent trade recently as investors keep watch of a deadly viral outbreak of SARS-CoV-2 (COVID-19). The virus has affected over 210 countries and territories worldwide and two international conveyances. It has stopped the world and its economy. Massacres in the health care system have impacted cross-border relationships by locking down countries, further slowing the economy. Increasing fears over the continued spread of COVID-19 have led to aberrant behaviors in the stock market (see Fig. 1), broadly impacting the global economy. The reaction to the virus spread is quite dominating as the recent fall in the oil

Corresponding author
Kumer Pial Das,
kumer.das@louisiana.edu

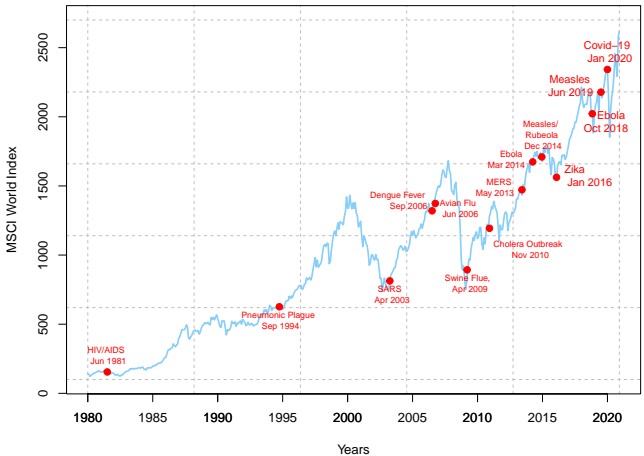

**Figure 1** **World epidemics and global stock market performance.**

price and stock composite indices around the world. *Baker et al. (2020)* have identified the COVID-19 pandemic as having the most significant impact on stock market volatility in the history of pandemics. After the shock, markets are tending to stabilize in recent days. According to economists and financial analysts, expecting a quick recovery from this volatile economic situation would be unrealistic. Economic and financial experts say the world economy will have to deal with COVID-19 for many years.

The statistical analysis of the stock market index is critically important to explore the impact of confirmed cases of COVID-19 on the overall stock price index. *Dey & Das (2022)* provided an analysis of the effect of the COVID-19 outbreak on the crude oil price. A very recent analysis revealed the volatility spillovers and co-movements among energy-producing, extracting, and transporting corporations' stock prices and evaluate how the COVID-19 pandemic creates negative WTI oil prices (*Corbet, Goodell & Günay, 2020*). A recent study by *Dey et al. (2021)* showed that COVID-19 cases and deaths, their local spread, and Google searches impact abnormal stock prices between January 2020 to May 2020. Understanding the market performance during the onset of deadly infectious diseases is important for many reasons.

Moreover, the COVID-19 pandemic has caused significant economic disruption, with stock markets worldwide experiencing sharp declines and volatility. The pandemic has created a new challenge for stock market forecasting models, as the infection rate and associated public health measures have become critical exogenous variables affecting market behavior. *Gupta et al. (2020)* used a vector autoregressive (VAR) model to examine the impact of COVID-19 on the stock market in India. They found that the infection rate was a significant predictor of stock market returns, with negative effects on both short- and long-term returns. The authors suggested that incorporating the infection rate into forecasting models could improve accuracy. Another recent study by *Ma & Yan (2022)* used a deep learning and artificial intelligence techniques to forecast the Shanghai Composite Index during the COVID-19 pandemic. The infection rate was included as an exogenous

variable in the model, and the authors found that it had a significant impact on stock market returns. The authors concluded that the hybrid model outperformed traditional models in forecasting accuracy. A similar study used a machine learning-based model to predict the stock market index in Taiwan during the Covid-19 pandemic (*Huang, Liu & Yao, 2020*). The authors included the infection rate and other exogenous variables in the model and found that they significantly improved forecasting accuracy. They suggested that including the infection rate in stock market forecasting models could help investors better understand the impact of the pandemic on the market. *Zaremba & Kizys (2021)* used wavelet coherence analysis to study the impact of Covid-19 on the US stock market and found evidence of significant linkages between the two. Similarly, *Chen et al. (2020)* used the Generalized AutoRegressive Conditional Heteroskedasticity (GARCH) model to analyze the volatility of the US stock market during the pandemic and found that the volatility increased significantly.

Furthermore, several other studies have also investigated the impact of the COVID-19 pandemic on the stock market using different modeling techniques. For example, *Ozer, Demir & Cosar (2021)* used daily stock prices and technical indicators data from 2015 to 2020, which includes both the pre-COVID-19 period and the COVID-19 period, to train and test the models. The results show that both random forest (RF) and deep neural network (DNN) models provide promising results in terms of forecasting accuracy and that the DNN model outperforms the RF model in terms of forecasting performance during the COVID-19 period. Similarly, another study used the ARIMA model to forecast the Karachi Stock Exchange (KSE) index during the pandemic period and found that the model accurately predicted the trend in the index (*Hasan, Rahman & Hasan, 2021*). *Alkhatib et al. (2022)* employed a structural time series-based model to forecast stock prices in the Gulf Cooperation Council (GCC) countries, such as Kuwait and Bahrain. Their findings indicated that the model yielded accurate forecasts, with Bahrain being the most affected country in this cohort due to the COVID-19 pandemic. They found that the model provided accurate forecasts. *Zaremba et al. (2020)* have focused on understanding the impact of COVID-19 on the US stock market volatility. None of these studies have focused on time series analysis to forecast stock indexes. Moreover, even though several studies have focused on a specific country's stock index, no attempt has been made to study world stock index, such as MSCI.

Thus, our primary focus is to employ time series analysis to predict future stock price indices concerning COVID-19 infection rates. We believe that the number of confirmed cases significantly impacts the stock index, and hence it will be considered a covariate in our analysis. In this research, the effect of covariates will be analyzed for S & P 500 stock Index data, MSCI World stock Index data, and CBOE volatility index (VIX) data. The description and details of the data are given the Section 'Methodology'. The data will be divided into a training set to train our model and a validation set to validate our model to see the model's performance on the test set. Finally, we will provide a prediction interval for the stock price index. The rest of the article is organized into four sections. The second section describes the data sets used in this study, the third section discusses the methods

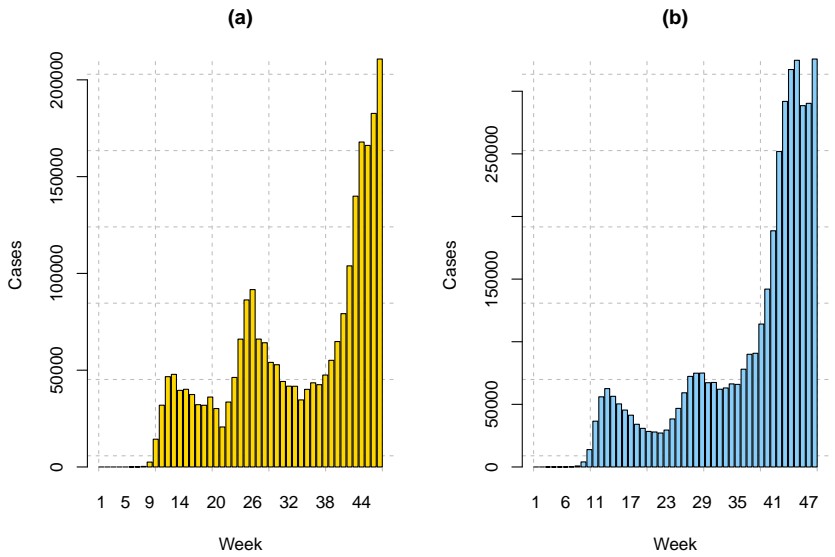

**Figure 2** COVID-19 weekly confirmed cases in the (A) USA and (B) twenty-three MSCI countries.

used, the fourth section checks stationarity assumptions, and we conclude with results and discussion in the fifth section.

## DATA DESCRIPTION

In this study, we have considered the weekly average of COVID-19 infection data for the USA and 23 MSCI Countries, including Hong Kong. The number of individuals infected from December 31, 2019, to February 12, 2021, has been included. December 31 has been chosen because the World Health Organization has confirmed and declared COVID-19 cases on this date. However, in the USA the first COVID-19 case was reported on January 21, 2020, and in the MSCI countries, the first COVID-19 case was reported on January 15, 2020. Therefore, for the USA, we have 56 weeks of data. Among these, we have used the first 47 weeks (January 21, 2020–December 11, 2020) data to train our model and the last nine weeks' data (December 12, 2020–February 12, 2021) to validate our model. The COVID-19 confirmed cases in the USA and twenty-three MSCI countries are displayed in Fig. 2. In this research, the following three stock indices have been considered.

### S & P 500 Index:

The S & P 500 index measures the stock performance of 500 large companies listed on stock exchanges in the United States. Many consider it one of the best representations of the US stock market. S & P 500 weekly index data from January 21, 2020, to February 12, 2021, has been analyzed in this study. We have considered weekly indices for the first 47 weeks (January 21, 2020–December 11, 2020) to train the model and weekly indices for the last nine weeks (December 12, 2020–February 12, 2021) to validate the trained model. The visualization of the training dataset for S & P 500 index is provided in Fig. 3A, and summary statistics of this dataset are presented in Table 1. It has been observed that the

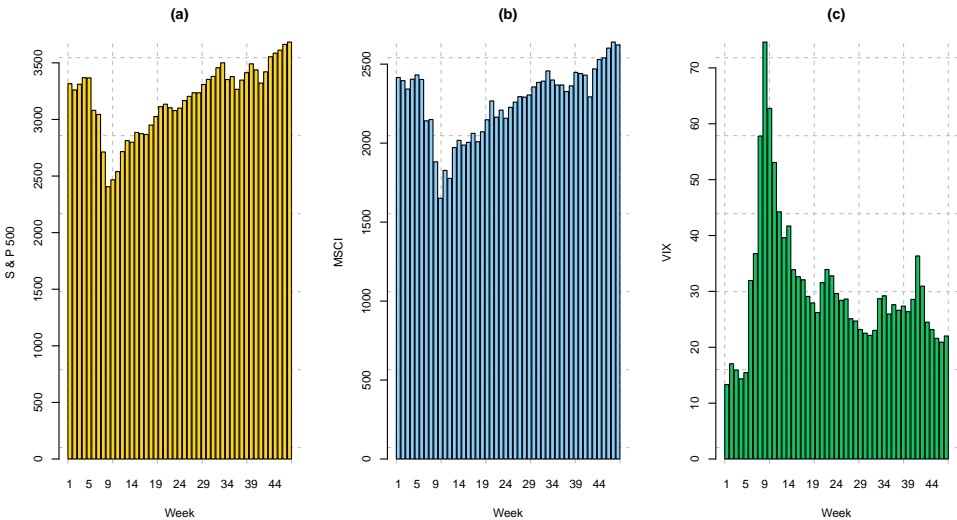

**Figure 3** Weekly price index for (A) S & P 500; (B) MSCI; and (C) VIX.

**Table 1** Summary statistics (sample size, minimum, first quartile, median, mean, standard deviation, third quartile, maximum, and kurtosis) of S & P 500, VIX, and MSCI.

| Data | $n$ | $Y_{min}$ | $Q_1$ | $Q_2$ | $\bar{Y}$ | SD | $Q_3$ | $Y_{max}$ | $\gamma$ |
|------|-----|-----------|-------|-------|-----------|-----|-------|-----------|----------|
| S & P 500 | 47 | 2406 | 3035 | 3260 | 3185 | 309.55 | 3379 | 3683 | 2.94 |
| VIX | 47 | 13.32 | 23.17 | 28.40 | 30.34 | 12.05 | 32.70 | 74.62 | 6.49 |
| MSCI | 48 | 1651 | 2146 | 2316 | 2264 | 224.22 | 2407 | 2640 | 3.05 |

sample range ($Y_{max} - Y_{min}$) and interquartile range ($Q_3 - Q_1$) are higher as compared to the pre-COVID time.

## MSCI world index:

The MSCI World Index is a market-cap-weighted stock market index of 1,643 stocks from companies across 23 developed countries worldwide. The US Canada, 15 European Countries, Australia, New Zealand, Israel, Japan, Hong Kong, and Singapore are included in this index. The index covers approximately 85% of each country's free float-adjusted market capitalization. This common benchmark for global stock funds is intended to represent a broad cross-section of global markets and is maintained by MSCI, formerly Morgan Stanley Capital International. The Weekly MSCI index data from January 15, 2020, to February 12, 2020, has been considered for this study. The MSCI indices for the first 48 weeks (January 15, 2020–December 11, 2020) have been considered training sets, and the indices for the last nine weeks (December 12, 2020–February 12, 2021) have been considered as the validation set. Like S & P 500, the visualization of the training dataset for the MSCI world index is provided in Fig. 3B, and summary statistics of this dataset are presented in Table 1. Similarly, as S & P 500 index, it is not surprising to observe that the sample range and interquartile range for the MSCI index are also higher compared to the normal time.

### CBOE volatility index (VIX):

The Chicago Board Options Exchange (CBOE) volatility index is a popular measure of the stock market's expectation of volatility based on S & P 500 index options. The VIX is often referred to as the fear index or fear gauge and is calculated and disseminated on a real-time basis by the CBOE. Portfolio managers and investors use the VIX to measure the level of risk, fear, or stress in the market when making investment decisions. The VIX index values move up when the market is falling. The reverse is true when the market advances. The data from January 21, 2020, to February 12, 2021, have been included in this study. The weekly VIX data from January 21, 2020, to December 11, 2020 (47 weeks) comprise the training set, and from December 12, 2020, to February 12, 2021, (9 weeks) include the validation set. Following the previous two indices, the visualization of the training dataset for VIX is provided in Fig. 3C, and summary statistics of this dataset are presented in Table 1. Trending with the previous two indices, it has been found that the sample range and interquartile range are higher compared to the normal time, but peaks and valleys in the data are more fluctuated than the previous two indices.

Moreover, to compare the spread of the three datasets, we compute the coefficient of variation (CV) $\left(= \frac{SD}{\bar{Y}} \times 100\%\right)$. We have found that the CVs for S & P 500, VIX, and MSCI are 9.72, 39.72, and 9.90 percent, respectively. These results indicate that VIX data has the highest spread from its sample mean, while that of S & P 500 has the least. Additionally, the last column of Table 1 presents the measures of kurtosis $(\gamma)$ for the three aforementioned datasets. Kurtosis is a measure of whether or not a data distribution is heavy- or light-tailed relative to a normal distribution. The measure of kurtosis for a normal distribution is 3. Since the values of kurtosis are very close to 3 (2.94 and 3.05) for S & P 500 and MSCI, these two datasets seem to be normally distributed without having many outliers or extreme observations as compared to the normal distribution. In contrast, the kurtosis measure for the VIX dataset is around 6.5 which indicates the data distribution is leptokurtic, which means the data is prone to produce more outliers or extreme observations than the normal distribution.

## METHODOLOGY

### Parametric forecasting methods

In the time series analysis, autoregressive moving average (ARMA) models were first introduced by *Whittle (1951)* and improved later by *Whittle (1963)* and *Whittle (1983)* to provide a parsimonious description of a stationary stochastic process in terms of two lower-order polynomials, one for the autoregressive (AR) part and the other for the moving average (MA) part (*Hannan, 1988*). But the models are also known as Box-Jenkins models (*Box & Genkins, 1970*) after the names of Box and Jenkins, who popularized the models. For a given time series, the ARMA model is one of the variants of Box-Jenkins model class which is a potent tool for understanding and predicting the future value of that series.

If the model includes AR terms of order $p$ and MA terms of order $q$ then the overall model is referred to as ARMA $(p, q)$. Formally, the process $\{Y_t\}$, t=0 $\{, \pm1, \pm2, \ldots\}$ is said to

be an ARMA$(p,q)$ process if $\{Y_t\}$ is stationary and if for every $t$ (*Brockwell & Davis, 2009*),

$$Y_t = \sum_{i=1}^{p} \phi_i Y_{t-i} + Z_t + \sum_{i=1}^{q} \theta_i Z_{t-i} \tag{1}$$

where $\phi_1, \phi_2, \ldots, \phi_p, \theta_1, \theta_2, \ldots, \theta_q$ are parameters, and $Z_t, Z_{t-1}, \ldots, Z_{t-q}$ are white noise error terms which follow $\{Z_t\} \sim WN(0, \gamma(h))$, where $\gamma(h) = \begin{cases} \sigma^2 & \text{if } h = 0 \\ 0 & \text{if } h \neq 0. \end{cases}$

Equation (1) can be written in a more compact form using a backward shift operator as follows:

$$Y_t - \sum_{i=1}^{p} \phi_i Y_{t-i} = Z_t + \sum_{i=1}^{q} \theta_i Z_{t-i} \Longrightarrow \phi(B) Y_t = \theta(B) Z_t, \qquad \forall t \tag{2}$$

where $\phi$ and $\theta$ are the $p$th and $q$th degree autoregressive and moving average polynomials, respectively in the above difference equations and are given by

$$\phi(x) = 1 - \phi_1 x - \phi_2 x^2 - \ldots - \phi_p x^p,$$

$$\theta(x) = 1 + \theta_1 x + \theta_2 x^2 + \ldots + \theta_q x^q$$

and B is the backward shift or lag operator defined as

$$B_j Y_t = Y_{t-j}, \quad j = 0, \pm 1, \pm 2, \ldots\ldots$$

Clearly, if $\theta(x) \equiv 1$ in Eq. (2), then the process

$$\phi(B) Y_t = Z_t, \qquad \forall t$$

is known as AR process of order $p$ and is symbolically denoted by AR$(p)$. Furthermore, if $\phi(x) \equiv 1$ in Eq. (2), then the process

$$Y_t = \theta(B) Z_t, \qquad \forall t$$

is known as MA process of order $q$ and is denoted by MA$(q)$.

ARMA models can be estimated by using the Box–Jenkins methodology, which is further divided into three major components.

- Identification: Identifying orders $p$ and $q$ for ARMA $(p,q)$
- Estimation: Estimating model parameters $\phi$s, $\theta$s, and $\sigma^2$.
- Diagnostics: Checking for overfitting and verifying the model assumptions using residual.

Since we wish to include covariate(s) in our analysis, we must incorporate the independent variables in ARMA$(p,q)$ model defined in Eq. (1). However, these models are uncommon and are known as autoregressive–moving-average with exogenous inputs model (ARMAX model). ARMAX model with $p$ autoregressive terms, $q$ moving average terms, and $r$ exogenous inputs terms is referred to as ARMAX$(p,q,r)$, which contains the AR$(p)$, MA$(q)$, and a linear combination of $r$ terms of known and external time series $X_t$. Thus, an ARMAX$(p,q,r)$ is given by *Brockwell & Davis (2009)*

$$Y_t = \sum_{i=1}^{p} \phi_i Y_{t-i} + Z_t + \sum_{i=1}^{q} \theta_i Z_{t-i} + \sum_{i=1}^{r} \beta_i X_{t-i} \tag{3}$$

where $\beta_1, \beta_2, \ldots, \beta_r$ are the parameters of exogenous input $X_t$.

## Random walk forecasting methods

The theory of random walks usually raises many challenging questions primarily because many "technical analysts" and "chartists" ask whether the random walk theory accurately describes reality. Indeed, the random walk approach is radically different from market analysis and starts from the premise that the stock exchanges are examples of efficient markets. In an efficient market, at any point in time, the actual price of a stock will be a reasonable estimate of its intrinsic value. The theory of random walk states that a series of stock price changes have no memory- the series' history can not be used to predict the future meaningfully. The future path of the price level is no more predictable than the path of a series of cumulated random numbers (*Fama, 1970*).

### Average method

The forecasts of all future values are equal to the average (or "mean") of the data at hand. If we let the existing data be denoted by $Y_1, Y_2, \ldots, Y_T$, then for forecast horizon $h$, forecasts for $Y_{T+h}$ are given by

$$\hat{Y}_{T+h|T} = \frac{\sum_{t=1}^{T} Y_t}{T} = \bar{Y}, \quad h \in \mathbb{Z}. \tag{4}$$

Here, $h$ is an integer such that $h \geq 1$.

### Naïve method

In the Naïve forecast, for any forecast horizon $h$, the forecast value will be the last observed value in the series.

$$\hat{Y}_{T+h|T} = Y_T, \quad h \in \mathbb{Z}. \tag{5}$$

This method dominates other methods in many situations in economic and financial time series. Since the forecast from a naïve approach is optimal when data follow a random walk, this method is also known as the random walk forecast method.

### Drift method

An alternative to the naïve method is to allow the forecasts to increase or decrease over time, where the amount of change over time (also known as drift) is set to be the average change in the data at hand. Thus, the forecast for horizon $h$ is $\hat{Y}_{T+h|T}$ and is given by:

$$\hat{Y}_{T+h|T} = Y_T + \frac{h}{T-1} \sum_{t=2}^{T} (Y_t - Y_{t-1}) = Y_T + h \left( \frac{Y_T - Y_1}{T-1} \right). \tag{6}$$

This method is equivalent to drawing a line between the first and last observations in the series and extrapolating it into the future.

## Tests for stationarity

It is important to check the stationarity of a series before fitting it to a model. In other words, it needs to be determined whether the time series is constant in mean and variance. We employ a couple of methods to check stationarity, as outlined below.

### Autocorrelation function (ACF)

The autocorrection function (ACF) test is a statistical method used to determine the presence of autocorrelation in a time series data set. It measures the correlation between a series and its lags, *i.e.*, the correlation between the data points separated by a given lag interval (*Venables & Ripley, 2002*). The mathematical formula for ACF is as follows:

$$ACF(l) = \frac{1}{n} \sum_{t=1}^{n} \frac{(Y_t - \bar{Y})(Y_{t-l} - \bar{Y})}{S^2}$$

where $l$ is the lag interval; $n$ is the number of observations in the time series; $Y_t$ is the value of the time series at time $t$; $\bar{Y}$ is the mean of the time series; and $S^2$ is the variance of the time series.

The ACF test is used to determine whether a time series is stationary or not. If the autocorrelation is zero or close to zero, the time series is stationary, and the ACF plot will resemble white noise. However, if the autocorrelation is high, then the time series is non-stationary, and the ACF plot will show a pattern of spikes or waves (*Box, Genkins & Reinsel, 2015*; *Brockwell & Davis, 2002*). The ACF test is widely used in econometrics, finance, and other fields to analyze time series data. It is a valuable tool for detecting trends, seasonal patterns, and other types of time series behavior (*Box, Genkins & Reinsel, 2015*).

### The Ljung–Box test

The Ljung–Box test is a standard method for model selection and is often used in time series analysis. The Ljung–Box test examines whether there is significant evidence for non-zero correlations at given lags, with the null hypothesis of independence or stationarity in a given time series (*Harvey, 1993*; *Ljung & Box, 1978*; *Box & Pierce, 1970*; *Brockwell & Davis, 2002*). The Ljung–Box test statistic is calculated as follows:

$$Q(k) = n(n+2) \sum_{l} \frac{r_l^2}{n-l}$$

where $n$ is the sample size, $k$ is the number of lags to consider, $r_l$ is the ACF at lag $l$, and $Q(k)$ is the test statistic which follows chi-squared distribution with $k$ degrees of freedom. A low $p$-value (*e.g.*, $p < 0.10$ or $0.05$) will indicate the non-stationarity of the series.

### Augmented Dickey–Fuller (ADF) test

Another common and familiar statistical method for stationarity in time series literature is the Augmented Dickey–Fuller (ADF) test used to test for the presence of a unit root in time series data (*Banerjee et al., 1993*; *Said & Dickey, 1984*). A unit root is a feature of a time series that indicates the presence of a stochastic trend. The ADF test helps determine if a time series is stationary or non-stationary. The mathematical formula for the ADF test is as follows:

$$\Delta Y_t = \rho Y_{t-1} + \delta_t + \epsilon_t,$$

where $\Delta Y_t$ is the first difference of the time series data; $\rho$ is the coefficient of the lagged dependent variable; $\delta_t$ is a constant term that includes any deterministic trends in the data, and $\epsilon_t$ is the error term. The null hypothesis of the ADF test is that the time series has a

unit root, meaning it is non-stationary (*Dickey & Fuller, 1979*). The alternative hypothesis is that the time series is stationary. The ADF test statistic is compared to a critical value based on the significance level and the sample size. If the test statistic is less than the critical value, the null hypothesis is rejected, and the time series is considered stationary.

The ADF test is commonly used in time series analysis to evaluate the stationarity of a time series and to determine the order of differencing required to make the time series stationary (*Stock & Watson, 1993*). If the time series is found to be non-stationary, it may be necessary to take first differences or higher order differences to make the time series stationary.

### Kwiatkowski-Phillips–Schmidt–Shin (KPSS) test

The KPSS test (*Kwiatkowski & Phillips, 1992*) is a statistical method used to test for the presence of a unit root in time series data. Unlike the ADF test, the KPSS test assumes that the null hypothesis is stationarity and the alternative hypothesis is non-stationarity. The mathematical formula for the KPSS test is as follows:

$$Y_t = \mu_t + \epsilon_t$$

where $Y_t$ is the time series data; $\mu_t$ is the deterministic trend function; and $\epsilon_t$ is the error term. The null hypothesis of the KPSS test is that the time series is stationary, and the alternative hypothesis is that it is non-stationary. The test statistic is calculated based on the sum of squared deviations from the estimated trend function. If the test statistic exceeds the critical value, the null hypothesis is rejected, and the time series is considered non-stationary. The KPSS test is commonly used in time series analysis to evaluate the stationarity of a time series and to determine if differencing is required to make the time series stationary.

## Test for randomness

In time series analysis, it is often a matter of interest to assess whether the series is a random walk or autocorrelated. To check this issue, we have several statistical hypothesis tests, namely, Wald-Wolfowitz Runs test (*Siegel & Castellan, 1988*; *Siegel, 1956*) and Bartels test (*Bartels, 1982*). Bartels test is typically more potent than the Runs test. Thus, we conclude the null hypothesis of the sequence generated by a random process versus the alternative hypothesis of the sequence generated by a process containing either persistence or frequent changes in direction using the Bartels test.

## Association analysis between prices (or indices) and COVID-19 cases

We aim to assess the feasibility of incorporating COVID-19-confirmed cases as a potential regressor into parametric analysis. To accommodate the number of COVID-19 cases into Box-Jenkin's methodology for forecasting prices or indices, it is recommended to assess the significance of the association between the number of weekly COVID-19 cases and weekly stock prices (or indices). Here, we have tested the significance of the Pearsonian product-moment correlation.

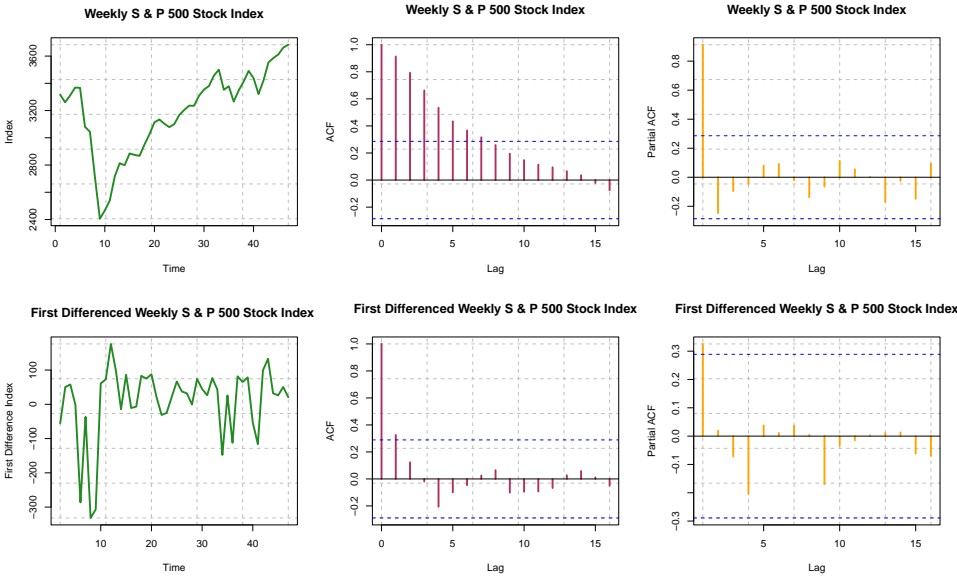

**Figure 4** **Statinarity assumptions of S & P 500 index data.**

# RESULTS AND DISCUSSION

## Stationarity assumptions

### For S & P 500 data

For the original series, ACF is not decaying fast for different time lags, so the series is visually non-stationary (see Fig. 4). In contrast, for the first difference of the series, ACF decays very quickly, which is indicative of the stationarity of the differenced series. Further, we must perform a statistical hypothesis test to substantiate the stationarity. We have performed the quantitative tests for testing stationarity by the Ljung–Box test. For the original series (S & P data), $p$-value $< 2.2 \times 10^{-16}$; and for the first difference series, the $p$-value is 0.4127. Thus, though the original series is a non-stationary series, the first difference series is stationary by the Ljung–Box test, and these outcomes are consistent with what we have seen from ACF plots. For the original series, the $p$-valu $e < 0.01$; for the first difference series, the $p$-value is 0.01967 from the ADF test. Thus, both the original series and the first difference series do not have unit roots. That is, both the original and the first difference series are stationary by the ADF test. KPSS test provides the $p$-value of 0.0642 for the original series, and that for the first difference series is greater than 0.10. The original series is not a trend stationary series, but the first difference series is indeed a trend stationary series by quantitative statistical hypotheses tests. Overall, we conclude that the original series of weekly S & P 500 stock indices are not stationary, but the first differences considered here are stationary.

### MSCI data

ACF is not decaying fast for different time lags for the original series, so the series is visually non-stationary (see Fig. 5). For the first difference of the series, ACF decays very quickly,

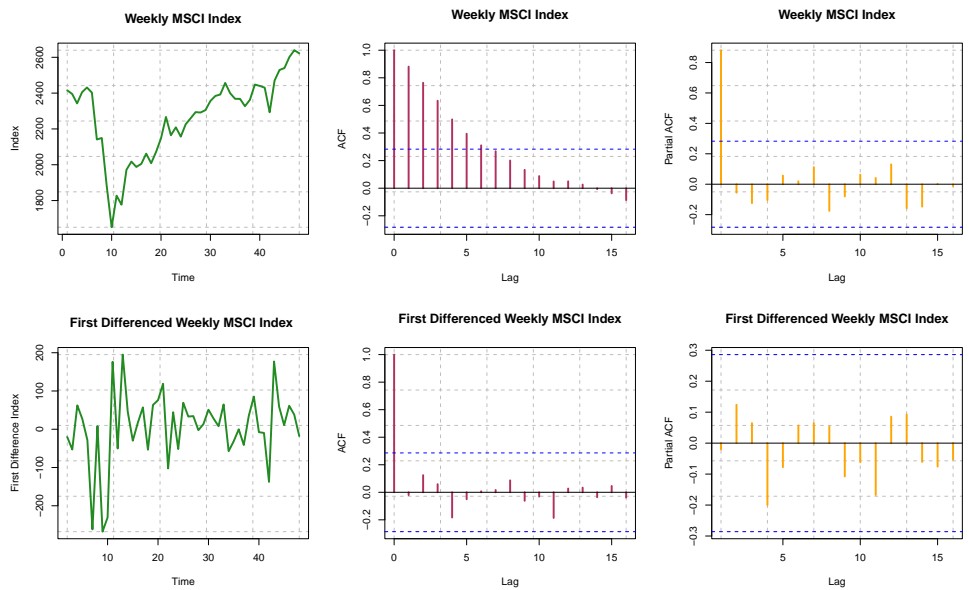

**Figure 5  Stationarity assumption of MSCI world index data.**

which indicates stationarity, but we need further statistical tests to confirm stationarity. For the original series, the Box-Ljung test provides a $p$-value of $< 2.2 \times 10^{-16}$; but for the first difference series, the $p$-value is 0.9605. Based on the results presented here, the original series is non-stationary, but the first difference series is stationary. For the original series, the $p$-value is 0.06524; for the first difference series, the $p$-value is 0.02559 from the ADF test. Thus, with the smaller nominal significance level ($\alpha = 0.05$), we may conclude that the first difference series for MSCI is stationary. Therefore, the original series seems to have unit roots, but the first difference series does not. For the original series, the $p$-value is 0.03821; for the first difference series, the $p$-value i $s > 0.1$ from the KPSS test. The difference series seems trend stationary, but the original series was not. Overall, we may conclude that the original series of MSCI stock indices considered here is not stationary, but the first difference of the series is found to be stationary.

### VIX data
ACF is not decaying for different time lags for the original series, so the series is visually non-stationary (see Fig. 6). In contrast, for the first difference of the series, ACF decays relatively faster, which is indicative of stationarity. Further, we must perform a statistical hypothesis test to substantiate the stationarity. For the original series, the $p$-value is $7.737 \times 10^{-16}$; for the first difference series, the $p$-value is 0.1808 from the Box-Ljung test. Consequently, though the original series is non-stationary, the first difference series is stationary. For the original series, the $p$-value is less than 0.01; for the first difference series, the $p$-value is 0.02095 from the ADF test. Hence, both the original and the first difference series are likely to be stationary. Again, $p$-values are more significant than 0.01 from the KPSS test for the original and first difference series. Both series are trend stationary. Overall,

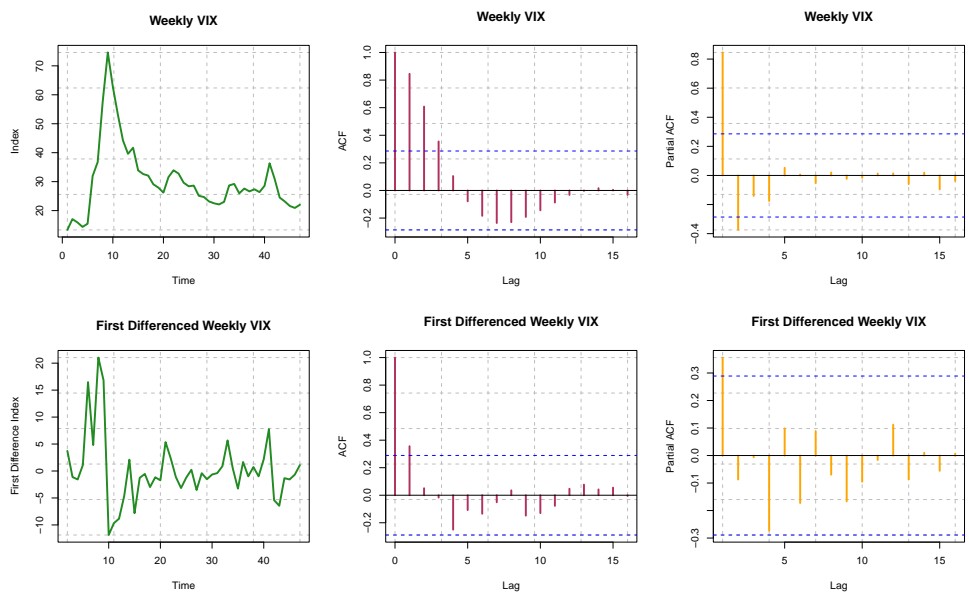

**Figure 6** Stationarity assumption of VIX data.

we may conclude that the original series of the VIX index seems to be trend stationary, but the first difference of the series is undoubtedly stationary.

## Randomness assumption
### S & P 500 data

For the original series, the $p$-value is $1.949 \times 10^{-10}$ indicating that the original series is not a random walk. Likewise, for the first difference series, the $p$-value is 0.09555, meaning that the first difference series is also not a random walk if the nominal significance level is 0.10. Thus, in this research, we can employ Box-Jenkin's methodology for the prediction of S & P 500 stock indices using the first difference series.

### MSCI data

For the original series, the $p$-value is $8.692 \times 10^{-10}$ indicating that the original series is not a random walk. In contrast, for the first difference series, the $p$-value of 0.4891 leaves the trace that the first difference series is a random walk.

### VIX data

For the original series, the $p$-value is $5.915 \times 10^{-09}$ indicates that the original series is not a random walk. Similarly, for the first difference series, the $p$-value of 0.04727 demonstrates that the first difference series is also not a random walk at a nominal significance level of 0.05.

However, when a time series is non-stationary, the general practice is to make the series difference stationary. Moreover, if the difference stationary series is not autocorrelated, the original series is a random walk. If so, any parametric time series modeling should be used for forecast purposes. In our preliminary analysis, we have found that the first difference

series of S & P and VIX are stationary and autocorrelated (not random walk). Still, the first difference series of MSCI is a stationary but random walk. Thus, we may employ a parametric method for price or index forecasting for the first difference between S & P 500 series and VIX, which is Box-Jenkin's methodology. On the other hand, we may deploy random walk forecasting methods for MSCI index forecasting.

## Association analysis

### S & P 500 weekly indices and weekly COVID-19 cases

Here, we have considered weekly data for both S & P 500 index and COVID-19 cases in the USA. The Pearsonian product-moment correlation between S & P 500 weekly index and Weekly COVID-19 cases in the USA is 0.5500 with a $p$-value of 6.204 $10^{-05}$. Therefore, the number of confirmed COVID-19 cases is significantly correlated with S & P 500 stock indices.

### Weekly VIX and weekly COVID-19 cases

Here, we have considered weekly data for VIX and COVID-19 cases in the USA. The Pearsonian product-moment correlation between VIX and Weekly COVID-19 cases in the USA is $-0.2487$ with a $p$-value of 0.09184. Therefore, the number of confirmed COVID-19 cases significantly correlates with VIX at a nominal significance level of 0.10.

## Forecast using Box-Jenkin's method

In forecasting prices or indices using Box-Jenkin's methodology for stationary time series or difference stationary time series, it is desirable to develop an appropriate order of autoregressive (AR) and moving average (MA) terms. In this research, we select the orders of AR and MA using the cross-validation method. This is one of the most useful statistical and machine learning methods in order selection.

We consider Akaike Information Criterion (AIC) (*Akaike, 1974*) as our model selection criterion, which is calculated by AIC = -2 log L + 2p, where L is the maximum value of the likelihood function of the model, and $p$ is the number of estimated parameters in the model. The AIC value is calculated based on the number of parameters used in the model and the log-likelihood function, which measures how well the model fits the data. A lower AIC value indicates a better fit of the model to the data.

We select the order of AR and MA that provide the model with the smallest AIC value. For each of the data, we present the order and AIC (see Table 2) and ACF plot (see Fig. 7) of residual of final models for S & P 500 and VIX. Detail guidelines for model selection can be found in *Hyndman & Khandakar (2008)* and *Wang, Smith & Hyndman (2006)*. Estimates of the corresponding model parameters and their test of significance have been presented in Table 3. The AR and MA parameters are highly significant for S & P 500 and VIX data, whereas the parameter for COVID-19 infection rate is somewhat significant for both the data.

### S & P 500 index data

In this study, we have considered the weekly number of COVID-19 confirmed cases as a regressor and weekly S & P 500 indices from December 12, 2020, to February 12, 2021

**Table 2    Order of ARIMA and AIC values of optimum models.**

| Data | ARIMA order | AIC |
|------|-------------|-----|
| S & P 500 | (1, 1, 0) | 557.01 |
| VIX | (2, 0, 1) | 300.49 |

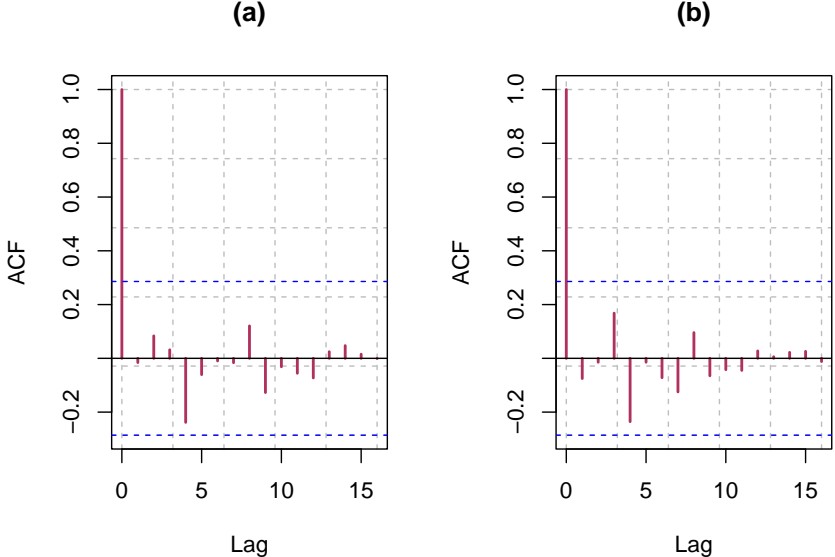

**Figure 7    (A) ACF plot of residuals from optimum model for S & P 500 index; (B) ACF plot of residuals from optimum model for VIX.**

**Table 3    Estimates of model parameters along with standard error (SE) and test of significance of model parameters.**

| Data | Model | Estimates | SE | *p*-value |
|------|-------|-----------|-----|-----------|
| S & P 500 | ARIMA(1, 1, 0) | $\hat{\phi}_1 = 0.324$ | 0.138 | 0.019 |
| | | $\hat{\beta}_1 = 0.125$ | 0.068 | 0.066 |
| VIX | ARIMA (2, 0, 1) | $\hat{\phi}_1 = 1.246$ | 0.130 | <0.000 |
| | | $\hat{\phi}_2 = -0.435$ | 0.136 | 0.001 |
| | | $\hat{\theta}_1 = 32.316$ | 7.116 | <0.000 |
| | | $\hat{\beta}_1 = 0.103$ | .061 | 0.091 |

(9 weeks) for forecasting using our optimum trained model. Forecast indices and 80%, 95%, and 99% Prediction intervals are presented in the following table (see Table 4). The visualization of these results has been presented in Fig. 8.

### *VIX data*

Similar to S & P 500, we have considered the weekly number of COVID-19 confirmed cases as a predictor variable and weekly VIX data from December 12, 2020, to February 12, 2021 (9 weeks) for forecasting using our optimum trained model. Forecast indices, along with 80%, 95%, and 99% prediction intervals, are presented in the following table (see Table 5).

**Table 4** Forecast for S & P 500 indices along with 80%, 95%, and 99% prediction intervals (PIs) from the ARIMAX method.

| Horizon | Forecast | 80% PI | 95% PI | 99% PI |
|---|---|---|---|---|
| 48 | 3681.536 | (3555.140, 3807.932) | (3488.230, 3874.842) | (3427.489, 3935.583) |
| 49 | 3649.608 | (3439.857, 3859.359) | (3328.822, 3970.394) | (3228.024, 4071.192) |
| 50 | 3656.062 | (3379.217, 3932.906) | (3232.665, 4079.458) | (3099.624, 4212.499) |
| 51 | 3727.503 | (3394.535, 4060.470) | (3218.273, 4236.732) | (3058.262, 4396.743) |
| 52 | 3704.052 | (3322.462, 4085.642) | (3120.461, 4287.644) | (2937.083, 4471.022) |
| 53 | 3610.386 | (3185.505, 4035.267) | (2960.586, 4260.186) | (2756.404, 4464.368) |
| 54 | 3563.579 | (3099.368, 4027.790) | (2853.629, 4273.529) | (2630.546, 4496.611) |
| 55 | 3510.924 | (3010.446, 4011.402) | (2745.509, 4276.339) | (2504.999, 4516.850) |
| 56 | 3459.464 | (2925.170, 3993.758) | (2642.332, 4276.596) | (2385.571, 4533.357) |

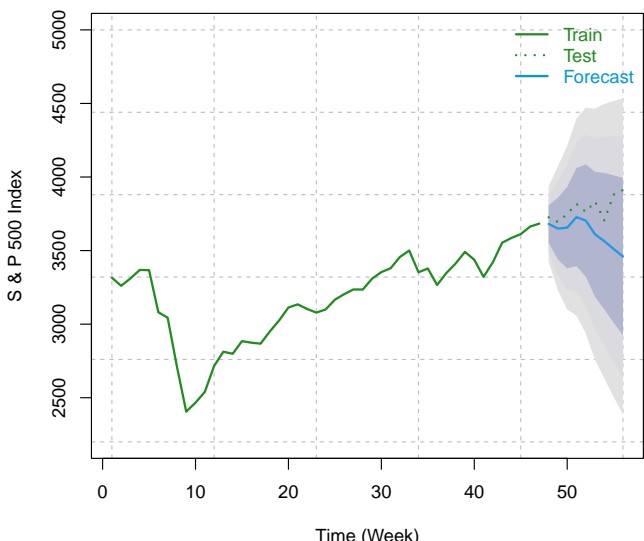

**Figure 8** Train series, test series, and forecast for S & P 500 index along with 80% (inner most), 95% (middle), and 99% (outer most) prediction bands.

These results have been displayed in Fig. 9 along with the original series. Since VIX is an index and it cannot take a negative value. To address this issue, we fit the model on natural logarithm-transformed data and, later on, exponentiated the results to bring them back to their original scale.

### MSCI data

Like S & P 500 Index, we have considered the weekly MSCI Index from December 12, 2020, to February 12, 2021 (9 weeks) for forecasting using our train model. Here, forecasts have been made using three different random walk forecasting methods: the mean method, the naïve method, and the drift method, as described in the fourth section. Forecast of the index along with 80%, 95%, and 99% Prediction intervals are presented in the following

**Table 5** Forecast for VIX along with 80%, 95%, and 99% prediction intervals (PIs) from ARIMAX method.

| Horizon | Forecast | 80% PI | 95% PI | 99% PI |
|---|---|---|---|---|
| 48 | 22.741 | (15.752, 29.731) | (12.051, 33.432) | (8.692, 36.791) |
| 49 | 23.642 | (12.472, 34.812) | (6.559, 40.725) | (1.191, 46.093) |
| 50 | 23.092 | (9.456, 36.727) | (2.238, 43.945) | (0.013, 50.498) |
| 51 | 20.565 | (5.685, 35.445) | (0.111, 43.322) | (0.000, 50.473) |
| 52 | 20.833 | (5.418, 36.248) | (0.064, 44.408) | (0.000, 51.815) |
| 53 | 23.233 | (7.629, 38.837) | (0.532, 47.098) | (0.000, 54.596) |
| 54 | 24.374 | (8.718, 40.030) | (0.431, 48.318) | (0.001, 55.841) |
| 55 | 25.769 | (10.105, 41.434) | (1.812, 49.727) | (0.003, 57.255) |
| 56 | 27.192 | (11.527, 42.857) | (3.234, 51.150) | (0.014, 58.678) |

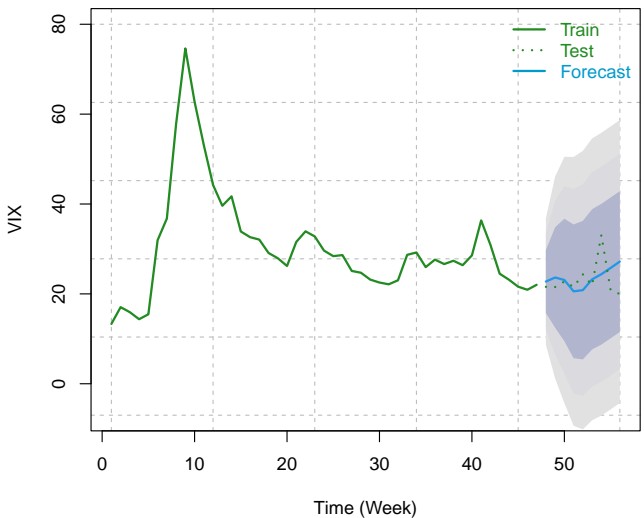

**Figure 9** Train series, test series, and forecast for VIX along with 80% (innermost), 95% (middle), and 99% (outermost) prediction bands.

tables (see Tables 6, 7, and 8) for the aforementioned methods. These results have been shown schematically in Fig. 10, along with the original series.

From the accuracy measures (*Hyndman & Athanasopoulos, 2018*; *Hyndman & Koehler, 2006*; *Armstrong, 1978*) presented in Table 9, it can be concluded that the best method for MSCI data forecasting, based on the RMSE and MPAE, is the drift method, which suggests that the trend is more important than the seasonality in this series.

## CONCLUDING REMARKS

Forecasting methodologies and modeling are always challenging due to strict assumptions behind the time series forecasting methods. Even assumptions are intrinsically strict for applying any parametric methods of forecasting. In this research, we have started with three different worldwide stock or stock-related indices, namely, S & P, MSCI,

**Table 6  Forecast for MSCI world indices along with 80%, 95%, and 99% prediction intervals (PIs) from the mean method.**

| Horizon | Forecast | 80% PI | 95% PI | 99% PI |
|---|---|---|---|---|
| 49 | 2264.435 | (1969.974, 2558.895) | (1808.698, 2720.172) | (1656.279, 2872.59) |
| 50 | 2264.435 | (1969.974, 2558.895) | (1808.698, 2720.172) | (1656.279, 2872.59) |
| 51 | 2264.435 | (1969.974, 2558.895) | (1808.698, 2720.172) | (1656.279, 2872.59) |
| 52 | 2264.435 | (1969.974, 2558.895) | (1808.698, 2720.172) | (1656.279, 2872.59) |
| 53 | 2264.435 | (1969.974, 2558.895) | (1808.698, 2720.172) | (1656.279, 2872.59) |
| 54 | 2264.435 | (1969.974, 2558.895) | (1808.698, 2720.172) | (1656.279, 2872.59) |
| 55 | 2264.435 | (1969.974, 2558.895) | (1808.698, 2720.172) | (1656.279, 2872.59) |
| 56 | 2264.435 | (1969.974, 2558.895) | (1808.698, 2720.172) | (1656.279, 2872.59) |
| 57 | 2264.435 | (1969.974, 2558.895) | (1808.698, 2720.172) | (1656.279 2872.59) |

**Table 7  Forecast for MSCI world indices along with 80%, 95%, and 99% prediction intervals (PIs) from the naïve method.**

| Horizon | Forecast | 80% PI | 95% PI | 99% PI |
|---|---|---|---|---|
| 49 | 2621.89 | (2502.025, 2741.755) | (2438.572, 2805.208) | (2380.969, 2862.811) |
| 50 | 2621.89 | (2452.375, 2791.405) | (2362.639, 2881.141) | (2281.176, 2962.604) |
| 51 | 2621.89 | (2414.277, 2829.503) | (2304.373, 2939.407) | (2204.602, 3039.178) |
| 52 | 2621.89 | (2382.159, 2861.621) | (2255.253, 2988.527) | (2140.048, 3103.732) |
| 53 | 2621.89 | (2353.863, 2889.917) | (2211.978, 3031.802) | (2083.174, 3160.606) |
| 54 | 2621.89 | (2328.281, 2915.499) | (2172.854, 3070.926) | (2031.756, 3212.024) |
| 55 | 2621.89 | (2304.756, 2939.024) | (2136.875, 3106.905) | (1984.473, 3259.307) |
| 56 | 2621.89 | (2282.859, 2960.921) | (2103.388, 3140.392) | (1940.462, 3303.318) |
| 57 | 2621.89 | (2262.294, 2981.486) | (2071.935, 3171.845) | (1899.127, 3344.653) |

**Table 8  Forecast for MSCI world indices along with 80%, 95%, and 99% prediction intervals (PIs) from the drift method.**

| Horizon | Forecast | 80% PI | 95% PI | 99% PI |
|---|---|---|---|---|
| 49 | 2626.281 | (2505.253, 2747.309) | (2441.185, 2811.377) | (2383.024, 2869.538) |
| 50 | 2630.672 | (2457.702, 2803.642) | (2366.137, 2895.207) | (2283.014, 2978.330) |
| 51 | 2635.063 | (2421.023, 2849.103) | (2307.718, 2962.409) | (2204.858, 3065.268) |
| 52 | 2639.454 | (2389.793, 2889.115) | (2257.631, 3021.278) | (2137.653, 3141.256) |
| 53 | 2643.845 | (2361.938, 2925.752) | (2212.706, 3074.985) | (2077.232, 3210.459) |
| 54 | 2648.236 | (2336.410, 2960.063) | (2171.339, 3125.134) | (2021.487, 3274.986) |
| 55 | 2652.627 | (2312.593, 2992.662) | (2132.590, 3172.665) | (1969.182, 3336.073) |
| 56 | 2657.019 | (2290.093, 3023.944) | (2095.855, 3218.182) | (1919.524, 3394.513) |
| 57 | 2661.410 | (2268.640, 3054.180) | (2060.720, 3262.100) | (1871.969, 3450.850) |

and VIX, for modeling their data to forecast the future indices in conjunction with the COVID-19 confirmed cases. Other challenges in this research are gathering, compiling, and manipulating stock indices data to align with COVID-19 confirmed cases because of discrepancies in reporting stock indices (5 days a week) and COVID-19 confirmed cases (7 days a week). For the datasets considered in this research, S & P and VIX data satisfied

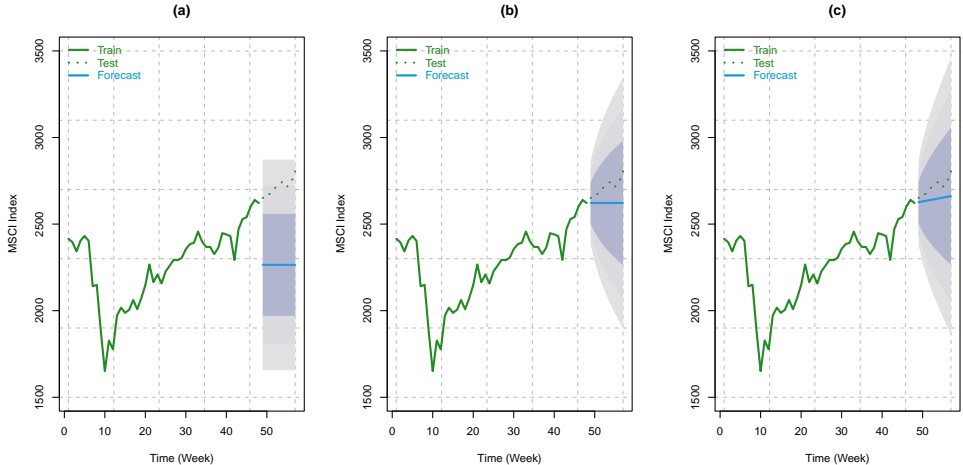

**Figure 10 Train series, test series, and forecast for MSCI index along with 80% (innermost), 95% (middle), and 99% (outermost) prediction bands.** (A) Using mean method; (B) using naïve method; (C) using drift method.

**Table 9 Accuracy measures (ME, mean error; RMSE, root mean squared error; MAE, mean absolute Error; MPE, mean percent error; MAPE, mean absolute percent Error) of forecast models for different datasets.**

| Data | Model | ME | RMSE | MAE | MPE | MAPE |
|------|-------|-----|------|-----|-----|------|
| S & P (Training) | ARIMA(1, 1, 1) | −0.340 | 95.428 | 68.961 | −0.050 | 2.269 |
| S & P (Test) | ARIMA(1, 1, 1) | 168.710 | 219.978 | 168.710 | 4.395 | 4.395 |
| VIX (Training) | ARIMA (2, 0, 1) | 0.255 | 5.217 | 3.512 | −1.930 | 11.770 |
| VIX (Test) | ARIMA (2, 0, 1) | −0.428 | 4.386 | 3.365 | −4.117 | 14.198 |
| MSCI (Training) | Mean | 0.000 | 221.867 | 179.527 | −1.061 | 8.325 |
| MSCI (Test) | Mean | 450.385 | 452.692 | 450.385 | 16.566 | 16.566 |
| MSCI (Training) | Naïve | 4.391 | 93.531 | 66.280 | 0.069 | 3.125 |
| MSCI (Test) | Naïve | 92.930 | 103.535 | 92.930 | 3.396 | 3.396 |
| MSCI (Training) | Drift | 0.000 | 93.428 | 65.626 | −0.127 | 3.099 |
| MSCI (Test) | Drift | 70.975 | 79.374 | 70.975 | 2.593 | 2.593 |

the assumptions for parametric forecasting methods. In contrast, MSCI data satisfied the assumptions for the random walk forecasting method.

It has been observed that the variant integrated ARMAX of Box-Jenkins parametric methods of forecasting for the S & P index and VIX does a good job of modeling the data. From Tables 4, 5 and Figs. 8, 9, it has been found that the forecasted indices are close to the original test set of data. In addition, the shortest of the three forecast intervals among 80%, 95%, and 99% contains the forecasted series, which is an indication that the adopted methodology performed well in capturing the underlying structure in the training data in connection with the COVID-19 confirmed cases which are further substantiated in the test data.

Nonstationarity data are not uncommon. Unlike S & P 500 and VIX, MSCI data showed nonstationarity behavior. One possible reason for such behavior could be due to the nature of the MSCI index, which spans over 23 countries throughout the world, and most likely has more noise than any other traditional index. Nonstationary data are challenging to model.

Nonetheless, random walk forecasting methods seem to perform a good modeling job in capturing the underlying structure in the training set of MSCI data substantiated by the test dataset. We have considered the mean, naïve, and drift methods of random walk forecasting. It has been found that in all the methods the forecasted indices are included by the 95% prediction intervals. However, for the naïve methods (see Tables 6, 7 and 8 and Fig. 10), the forecasted series is even closer to the original series and are also contained by the shortest prediction intervals. Since no study investigated the impact of COVID-19 infection rates on stock indices such as MSCI, no comparative analysis has been performed.

As the three indices investigated in this study are from different parts of the world, it was challenging to obtain uniform data as different countries have different holiday calendars for their stock markets. Moreover, as the reporting of the COVID-19 infection data varied from country to country significantly, it can be considered a limitation of the study.

## ACKNOWLEDGEMENTS

The authors are very grateful to the editorial members and the reviewers for their meaningful feedback, which helped us to improve the quality of the manuscript.

### Funding

The authors received no funding for this work.

### Competing Interests

Kumer P. Das is an Academic Editor for PeerJ.

### Author Contributions

- Mohammad Saha A. Patwary performed the experiments, analyzed the data, performed the computation work, prepared figures and/or tables, and approved the final draft.
- Kumer Pial Das conceived and designed the experiments, analyzed the data, authored or reviewed drafts of the article, and approved the final draft.

### Data Deposition

Code and raw data are available Supplemental Files.

### Supplemental Information

Supplemental information for this article can be found online at http://dx.doi.org/10.7717/peerj-cs.1532#supplemental-information.

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
