# Peer review of "Forecasting stock indices with the COVID-19 infection rate as an exogenous variable"

_PeerJ Computer Science, doi:10.7717/peerj-cs.1532_

## Round 0.1 · original submission · Major Revisions

Revise the manuscript as per the reviewers' comments.

Reviewer 1 ·

Basic reporting

The manuscript addresses a topic of extreme relevance to public health and economic. Totally, the present article is well-established and the subject is interesting, but some major revision should be considered. For example, the references are not cited properly, and there are relatively few references from the past three years. There are several typos and errors in the current manuscript, which need to be carefully checked and corrected.

Experimental design

To highlight the main contribution of the proposed study, a comparison on the numerical results of related works and the authors' method should be provided. I suggest that the authors should compare the results of the present work with some similar studies which have been done in the recent three years. Besides, much more explanations and interpretations must be added for the results, which are not enough in the current manuscript.

Validity of the findings

The scientific contribution of the current paper is doubtful. There is neither discussion nor demonstration on the advantage/innovation points of the proposed approach over other existed methods in the related works. The main contributions of this paper should be clarified by comparing with the existing results. Moreover, it is better to conduct more analysis and comparison on the correlation of respective numerical results.

Additional comments

I recommend the authors to perform a more detailed literature review, and position their paper compared to other related works in the literature. The authors should provide a (numerical) comparison between the proposed approach and the other well-known methods. Besides, the authors should also provide more detail descriptions for the implementation of the proposed approach.

Annotated reviews are not available for download in order to protect the identity of reviewers who chose to remain anonymous.

Reviewer 2 ·

Basic reporting

Very important

Experimental design

Good

Validity of the findings

Very important

·

Basic reporting

no comment.

Experimental design

no comment.

Validity of the findings

1. How do you define the "Training set to train the model and a validation the model "? Did you run any simulations to confirm the set up?
2. How do you justify the high SE (7.116) for theta1.hat in Table 1?

Additional comments

1. Add a sentence or two about your findings in the abstract.

---

## Round 0.2 · Minor Revisions

Statistical analysis is to be included to show the novelty of the paper. Further, include comparative analysis using visual aids to show the performance of the proposed approach.

Reviewer 1 ·

Basic reporting

This paper carries some interesting ideas, and their results are sound. Besides, the quality of the revised version of this submission has been improved according to reviewers’ comments. However, there are still some points that need to be improved.

Experimental design

I recommend the authors to make a complete statistical analysis and to include a series of indicators and tests such as Kurtosis, probabilities, standard deviation, etc. Besides, the number of observations taken in the sample, sampling mechanism and controls, should be well described.

Validity of the findings

It is important that the authors should present the correlation matrix and the covariance matrix. In addition, the authors should explain their dataset and results obtained via a visualization analysis.

Additional comments

I suggest the authors to deal with the above-mentioned comments for improving even more your paper. Therefore, I recommend to accept this submission after minor modifications.

·

Basic reporting

no comment

Experimental design

no comment

Validity of the findings

no comment

---

## Round 0.3 · accepted · Accept

Based on the reviewer's suggestion, I would recommend this paper for consideration for publication.

Reviewer 1 ·

Basic reporting

The quality of the paper has been significantly improved in this revision of the submission.

Experimental design

The scientific article presents originality, and the methods used are adequately described.

Validity of the findings

The presentation of the results is understandable, and the graphic display is satisfactory.

Additional comments

I believe it is ready for publication after minor corrections in this version.